# High-grade B-cell Lymphoma with *MYC* and *BCL2* Rearrangement Arising from Follicular Lymphoma: Presentation as a Large Peripancreatic Mass

**DOI:** 10.3390/diagnostics10030157

**Published:** 2020-03-14

**Authors:** Anna Shestakova, Sherif Rezk, Dara Ghasemizadeh, Ali Nael, Xiaohui Zhao

**Affiliations:** 1Department of Pathology, University of California, Irvine, CA 92868, USA; gshestak@uci.edu (A.S.); srezk@hs.uci.edu (S.R.); ANaelAmzajerdi@choc.org (A.N.); 2Department of Radiological Sciences, University of California, Irvine, CA 92868, USA; dghasemi@hs.uci.edu

**Keywords:** double-hit lymphoma 1, follicular lymphoma 2, CT scan 3

## Abstract

Follicular lymphoma, the second most common non-Hodgkin lymphoma (NHL), primarily affects adults and shows an indolent clinical course. Rare cases of follicular lymphoma transform to a high-grade B-cell lymphoma with *MYC* and *BCL2* rearrangements or “double-hit lymphoma”. Transformation to a “double-hit lymphoma” portends a worse prognosis and requires aggressive treatment. We report a comprehensive clinical, pathologic and radiographic review of a patient with previously undiagnosed low-grade follicular lymphoma that transformed into a “double-hit lymphoma”. The patient presented with a large heterogeneous mass 16 x 19 cm involving pancreatic head and neck and a mildly enlarged inguinal lymph node. Positron emission tomography (PET) study demonstrated Fluorodeoxyglucose (^18^F) (FDG)-avid peripancreatic mass. Tissue biopsy demonstrated a high-grade B-cell lymphoma with rearrangements t(14;18) and *MYC*, leading to the diagnosis of high-grade B-cell lymphoma with *MYC* and *BCL2* rearrangements. Excisional biopsy of an inguinal lymph node demonstrated low-grade follicular lymphoma. Clonality studies demonstrated the same immunoglobulin clone V7-4 in inguinal lymph node and peripancreatic mass. Therefore, diagnosis of a high-grade B-cell lymphoma with *MYC* and *BCL2* rearrangements that transformed from a low-grade follicular lymphoma was rendered. It is ultimately important to establish a tissue-based diagnosis at the different sites that are involved with lymphoma. Patient proceeded with the aggressive treatment with dose-adjusted etoposide, prednisone, vincristine, cyclophosphamide, doxorubicin and rituximab (EPOCH-R) treatment.

## 1. Introduction

Follicular lymphoma is the second most common non-Hodgkin lymphoma and it accounts for approximately 20% of all non-Hodgkin lymphoma [1]. The estimated incidence of follicular lymphoma in the United States is 3.18 cases per 100,000 people and it constitutes 35% of all non-Hodgkin lymphoma [2]. Incidence of follicular lymphoma in Europe 2.18 per 100,000 people [3]. Although follicular lymphoma affects all races, its incidence in Whites is double of that in Asians and Blacks. Incidence of follicular lymphoma rises with age and the median age of diagnosis is 65 years [1]. Patients usually present with painless peripheral and central lymphadenopathy. Staging studies usually demonstrate splenomegaly, liver and bone marrow involvement in 40, 50 and up to 80 percent of cases, respectively [4]. Despite the widespread disease, B-symptoms, such as fever and weight loss, are uncommon. Radiographically, CT and MRI are useful tools in assessing the extent of the disease. FDG-PET studies are instrumental in identifying patients with high risk of transformation [5].

Histological sections of the lymph nodes involved by follicular lymphoma show a follicular growth pattern, with tightly arranged neoplastic follicles that efface normal lymph node architecture [6]. Neoplastic follicles are composed of randomly admixed centrocytes and centroblasts and lack tingible body macrophages. Enumeration of centroblasts is used to determine the tumor grade, which has therapeutic implications. It is imperative to distinguish high-grade (grade 3A and 3B) from low-grade (grade 1 and 2), since therapy is guided by the grade of follicular lymphoma. Grade 1 and 2 follicular lymphoma contain 0–5 and 5–15 centroblasts per high-power field, respectively. Grade 3 lesions contain more than 15 centroblasts, but centrocytes are present in Grade 3A and absent in Grade 3B. Grade 3 follicular lymphoma is usually managed similarly to high-grade B-cell lymphoma. Follicular lymphoma cells are neoplastic B-cells that have a germinal center origin, and express B-cell markers (CD19, CD20, CD79a) and germinal center markers (Bcl6 and CD10). Ninety percent of grade 1–2 follicular lymphomas harbor a t(14;18)(q32;q21) translocation that results in the fusion of the anti-apoptotic gene *BCL2* on chromosome 18 to heavy-chain IGH on chromosome 14 (*BCL2/IGH*) [7]. Therefore, the majority of low-grade follicular lymphoma cells overexpress Bcl2, which is not expressed in reactive germinal centers. Overexpression of anti-apoptotic Bcl2 protein results in the persistence of the neoplastic clone. 

Histologic transformation or transformation of follicular lymphoma to a clinically aggressive subtype of lymphoma (commonly diffuse large B-cell lymphoma) occurs at approximately 2–3 percent per year [8]. A recent large prospective National Lympho Care Study (NLCS) of 2700 patients reported the 5- and 8-year transformation rates to be 13 and 19 percent, respectively [9]. At the time of transformation, approximately 25% of cases meet the criteria for high-grade B cell lymphoma with MYC and BCL-2 and/or BCL-6 rearrangements (“double-hit” lymphoma) [7]. Follicular lymphoma that transforms to a “double-hit lymphoma” is characterized by the *MYC* aberrations, including rearrangements and amplifications [10]. Progression to a high-grade B-cell lymphoma is associated with increased lymphadenopathy, the infiltration of extranodal sites, and the development of systemic symptoms [11]. The gold standard to diagnose transformation requires a biopsy from the involved lymph nodes or extranodal tissue that demonstrates sheets of large tumor cells and loss of the follicular architecture. These tumors resemble diffuse large B-cell lymphoma (DLBCL) or other aggressive lymphomas, and have high proliferative index, as indicated by Ki-67 immunostaining. Transformation to “double-hit lymphoma” warrants more aggressive treatment in comparison to transformation of DLBCL [11]. Therefore, it is imperative to perform Fluorescent in situ Hybridization (FISH) studies to determine rearrangements of *MYC*, in addition to t(14;18) or *BCL2/IGH*.

We report a comprehensive clinical, pathologic including molecular clonality studies and a radiologic review of a case of low-grade follicular lymphoma that progressed to a so-called “double-hit lymphoma” with *BCL2/IGH* and *MYC* rearrangements. Double-hit lymphoma was identified in the peripancreatic mass and demonstrated t(14;18) or *BCL2/IGH,* and rearrangement of MYC 8q32. Low-grade follicular lymphoma was diagnosed based on the excisional biopsy of an inguinal lymph node. In addition, molecular clonality studies confirmed the same origin of the low-grade and high-grade components of lymphoma. The next-generation sequencing of the immunoglobulin variable region determined the same clone, V7-4, in low-grade follicular lymphoma involving the inguinal lymph node and high-grade lymphoma with *BCL2/IGH* and *MYC* rearrangements involving the peripancreatic region. In order to establish accurate diagnosis, it is important to biopsy different sites that are involved with the lymphoma.

## 2. Case Presentation

A 57-year-old Asian female, with a past medical history of hypertension, hypercholesterolemia and pre-diabetes, presented with abdominal distension and acute right upper quadrant pain. She was in her usual state of good health until two month ago, when she developed B symptoms to include fatigue, fever, night sweats, chills and abdominal distension. Her symptoms progressed and included 8-pound weight loss, abdominal distension and pain that required evaluation through the Emergency Department. On physical examination, patient appeared visibly jaundiced, with a palpable centrally located abdominal mass. Labs showed mild moderate hypochromic anemia and mild leukocytosis. CT, with contrast of the abdomen and pelvis, revealed a large lobulated and heterogeneously enhancing mass, measuring 16 × 19 cm, involving the pancreatic head and uncinate region. There was also bulky lymphadenopathy encasing the superior mesenteric vessels (Figure 1 A–C). A PET scan showed FDG avid peripancreatic mass and superior mesenteric lymphadenopathy (Figure 1D) This case falls under category 4 of 45 CFR 46.101(b) Categories of Exempt Human Subjects Research. Research involving the collection or study of existing data, documents, records, pathological specimens or diagnostic specimens, if these sources are publicly available or the information is recorded by the investigator in such a manner that the subjects cannot be identified directly or through identifiers linked to the subjects. Furthermore, we do not require a signed authorization for case reports when there is not attached identifiable information: A HIPAA Authorization from the subject(s) of the Case Report is not required if there is access to PHI however there is no disclosure of PHI (outside of the covered/hybrid entity) in the Case Report publication.

In addition, a mildly enlarged 1.2 cm right inguinal lymph node was identified. The patient proceeded with an endoscopic ultrasound-guided fine needle aspiration (EUS-FNA) of the peripancreatic mass and ERCP with stent placement. The histopathologic evaluation revealed sheets of large atypical lymphocytes with hyperchromatic nuclei, irregular nuclear contours and scant cytoplasm (Figure 2A). 

Architectural pattern could not be appreciated due to the limitations of the FNA biopsy. Immunophenotypic studies revealed that atypical lymphocytes are neoplastic B-cells that express surface kappa light chain (by flow cytometry), CD20, CD10, Bcl-2, Bcl-6 and c-myc (Figure 2B–E). Neoplastic B-cells were negative for MUM-1, TdT and CD3. Epstein-Barr virus (EBV) was negative by EBV-encoded small RNA (EBER) in situ hybridization. Proliferative index, as judged by Ki-67 immunostain, was increased and estimated to be positive in 85% of cells (Figure 2F). Corresponding flow cytometry revealed sixty percent of kappa-restricted lymphoma cells. In this setting, differential diagnosis with reactive B-cell proliferation is not feasible due to the presence of sheets of highly atypical B-cells that showed surface kappa-light chain restriction, and an abnormal immunophenotype profile. Reactive B-cells express CD20, are expected to be smaller, and are usually admixed with T-cells. Germinal cell type non-neoplastic B-cells, in addition to CD20, express CD10 and Bcl6, but do not express Bcl2 and c-myc. Finally, a high proliferative index, estimated by Ki-67 at 85% is indicative of a high-grade B-cell neoplasm. Overall, histomorphological and immunophenotypic findings were of DLBCL, germinal center type. Fluorescence *in situ* hybridization (FISH) analysis was performed using probes specific to rearrangements involving *BCL6, MYC* and t(14;18), which are reported in high-grade/large B-cell lymphomas. The interphase study on the formalin-fixed paraffin-embedded tissue revealed a break apart signal for *MYC* (86% of cells, normal <11.6%) and dual fusion signal in the 14;18 (*IGH/BCL2*) probe set (90% of cells, normal <5.8%). The high-risk “double-hit” lymphomas are characterized by *MYC* translocation combined with *BCL2* and/or *BCL6* translocation. In the appropriate histomorphological setting, WHO 2016 recommends rendering diagnosis of a High-grade B-cell lymphoma with *MYC* and *BCL2* rearrangements, also known as “double-hit lymphoma”. Double-hit lymphomas are aggressive and require a more aggressive treatment than diffuse large B-cell lymphoma, including an aggressive treatment with chemotherapy and immunotherapy [12].

Translocation 14;18 (*IGH/BCL2*), although not specific to follicular lymphoma, raised the possibility of underlying follicular lymphoma that progressed to a high-grade B-cell lymphoma. Underlying follicular lymphoma can be unequivocally assessed using core needle or an excisional biopsy of the lymph node. Therefore, an excisional biopsy of the inguinal lymph node was performed. Gross examination revealed that the lymph node had a vaguely nodular tan-white cut surface. Histological sections revealed a follicular growth pattern characterized by tightly arranged follicles with attenuated mantle zones that effaced lymph node architecture (Figure 3A–B). Neoplastic germinal centers showed predominantly centrocytes with admixed centroblasts, loss of polarity and decreased tingible body macrophages. The neoplastic follicle center B-cells expressed surface kappa light chain, CD20, CD10, Bcl-2 and Bcl6 (not shown) (Figure 3 C–F). The follicular growth pattern was also highlighted by the CD21 immunostain that highlighted follicular dendritic cells (Figure 3G). The proliferative index was low, as judged by Ki-67 immunostaining (Figure 3H). Overall findings are of low-grade follicular lymphoma, of a predominantly follicular growth pattern, without evidence of transformation to high-grade B-cell lymphoma. Fluorescence in situ hybridization (FISH) analysis, using probes specific for rearrangements involving *BCL6, MYC* and t(14;18), revealed a dual fusion signal in the 14;18 (*IGH/BCL2*) probe (26% of cells, normal <5.8%) indicative of *IGH/BCL2* fusion. Overall, histomorphological, immunophenotypic and cytogenetic studies consistent with low-grade follicular lymphoma, grade 1–2 (Figure 3).

In addition, clonality studies using next-generation sequencing of the variable region of the immunoglobulin gene identified an identical clone, V7-4, in low-grade follicular lymphoma and high-grade B-cell lymphoma with *BCL2* and *MYC* rearrangements. Therefore, diagnosis of a high-grade B-cell lymphoma with *MYC* and *BCL2* rearrangements that transformed from a low-grade follicular lymphoma was rendered. The patient proceeded with the recommended dose-adjusted etoposide, prednisone, vincristine, cyclophosphamide, doxorubicin and rituximab (DA-EPOCH-R) treatment. CT of abdomen and pelvis with contrast post cycle 1 DA-EPOCH demonstrated tumor necrosis and a decrease in the size of peripancreatic mass to 17 × 19 cm (Figure 4).

The patient proceeded to receive cycle 1 of DA-EPOCH-R for induction, cycle 2, 3 and 4 DA-EPOCH-R. Bone marrow and cerebrospinal fluid were not involved with lymphoma. Overall, patient responded well to therapy. Her post-induction course was complicated by culture-negative neutropenic fever and headaches.

## 3. Discussion

Follicular lymphoma is one of the most common non-Hodgkin lymphomas that affects all races worldwide. Interestingly, incidence in Whites is twice than in Asians and Blacks [1,13]. As of today, risk factors for developing follicular lymphoma, appear to be elusive, with some evidence to suggest infectious etiology, drugs, toxins and genetic predisposition [4]. The vast majority of follicular lymphomas have the t(14;18), which results in the overexpression of B cell leukemia/lymphoma 2 (*BCL2*), an oncogene that blocks programmed cell death (apoptosis), leading to prolonged cell survival [14]. Over the course of the disease, approximately twenty percent of patients undergo progression to high-grade B-cell lymphoma [9,15]. The estimated annual rate of disease transformation is approximately 2–3 percent [2].

It is hypothesized that, initially, follicular lymphoma is comprised of multiple clones that have an indolent behavior. Transformation occurs through the clonal evolution where the most aggressive clone undergoes expansion [5,16]. Most commonly, follicular lymphoma transforms into a DLBCL [17]. The initial pathologic evaluation of DLBCL includes a stratification based on the immunohistochemical profile into germinal center B-cell subtypeand and non-germinal center B–cell subtype lymphomas. This classification is necessary to predict prognosis, outcomes and guide treatment. Approximately twenty-five percent of transformed follicular lymphoma acquire MYC rearrangement and transform to high-grade B cell lymphoma with *MYC* and *BCL2* rearrangements [10]. Transformation to a “double-hit” lymphoma has been well documented and can present a diagnostic challenge [18,19,20]. Patients that transform to “double-hit” lymphoma usually require dose-adjusted etoposide, prednisone, vincristine, cyclophosphamide, doxorubicin and rituximab (EPOCH-R) [13,14,15]. An aggressive EPOCH-R regimen can be potentially curative in the instances of de novo “double-hit” lymphomas. However, it is not effective in eradicating an underlying low-grade follicular lymphoma. While the major cause of relapse is due to the high-grade component, in patients with the transformed follicular lymphoma, follicular lymphoma recurrence can occur [21]. In addition, patients with underlying low-grade lymphoma are more likely to be resistant to Rituximab and radioimmunotherapy, due to the possible prior treatment of the low-grade component [21,22].

It is imperative to suspect a high-grade lymphoma in a patient who presents with B-symptoms and a large abdominal mass. Obtaining tissue from the most aggressive-appearing site is crucial to establish a diagnosis of a high-grade lymphoma. Comparison of tissue from the other site might be instrumental in identifying a low-grade component of the lymphoma. FISH studies targeting rearrangements of *MYC, BCL2* and *BCL6* are required to diagnose a double-hit lymphoma. Molecular clonality studies might be instrumental to establish the same origin in cases of high-grade B-cell lymphoma that do not show a concurrent low-grade component.

## Figures and Tables

**Figure 1 diagnostics-10-00157-f001:**
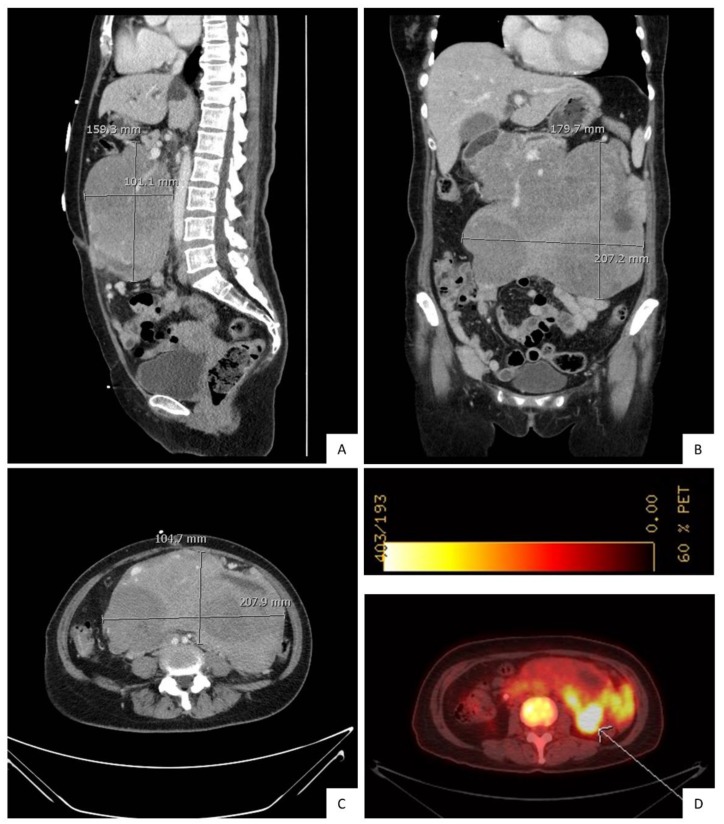
Initial abdominal and pelvic CT with contrast, prior to treatment. (**A**). sagittal, (**B**). coronal, and (**C**). transverse planes show a large lobulated heterogeneous mass involving the head of the pancreas and retroperitoneum. It measures approximately 19 x 22 cm (craniocaudal x transverse). (**D**). PET-FDG study demonstrates peripancreatic heterogeneous FDG-avid mesenteric mass (SUV 6.0).

**Figure 2 diagnostics-10-00157-f002:**
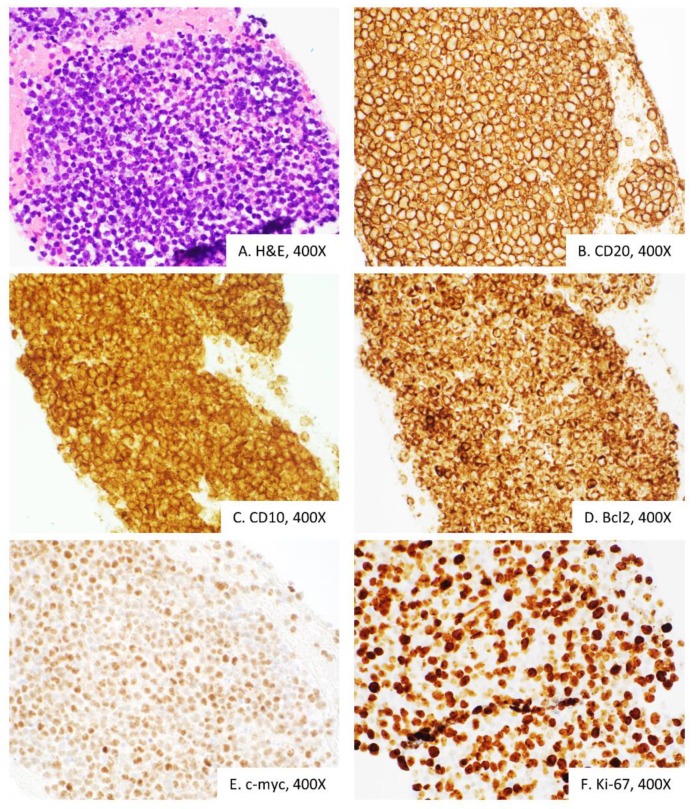
Endoscopic ultrasound-guided fine needle aspiration of the peripancreatic mass demonstrates sheets of large lymphocytes (**A**), that express B-cell marker CD20 (**B**), follicular center CD10 (**C**), anti-apoptotic protein Bcl2 (**D**), c-myc oncoprotein (**E**) and high levels of Ki-67 proliferative marker (**F**). Overall, histomorphologically and immunophenotypically consistent with diffuse large B-cell lymphoma.

**Figure 3 diagnostics-10-00157-f003:**
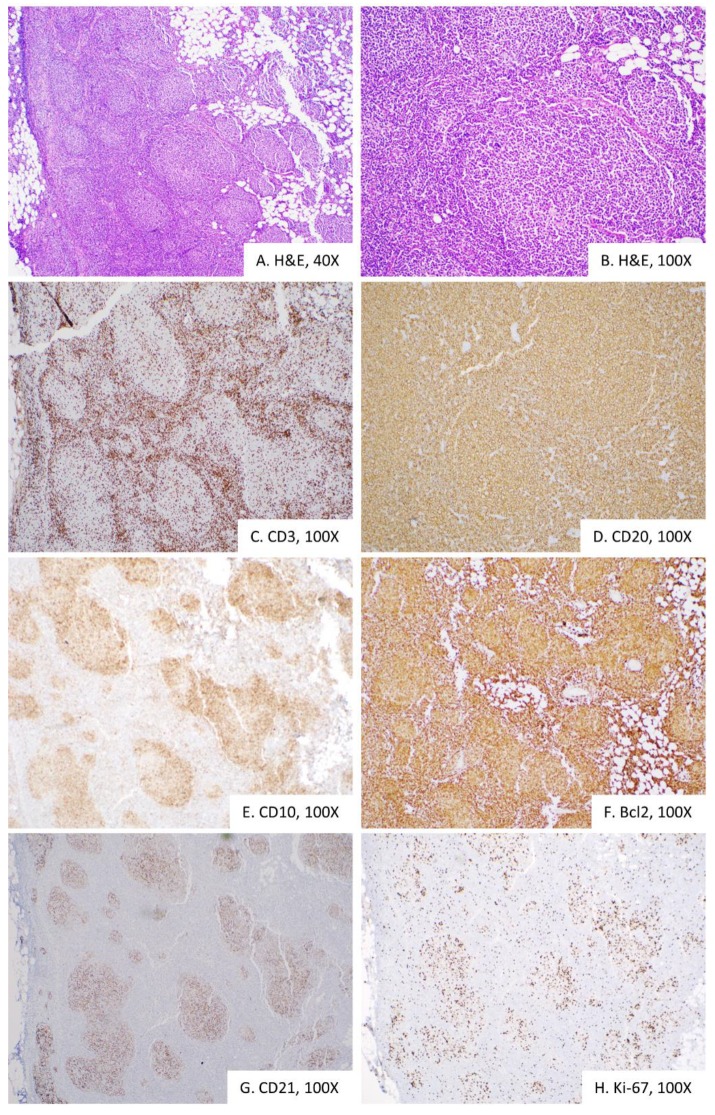
Excisional biopsy of the inguinal Endoscopic ultrasound-guided fine-needle aspiration of the peripancreatic mass demonstrates sheets of large lymphocytes (**A**), that express B-cell marker CD20 (**B**), follicle center CD10 (**C**), anti-apoptotic protein Bcl2 (**D**), c-myc oncoprotein (**E**) and high levels of Ki-67 proliferative marker (**F**). Overall, histomorphologically and immunophenotypically consistent with diffuse large B-cell lymphoma.

**Figure 4 diagnostics-10-00157-f004:**
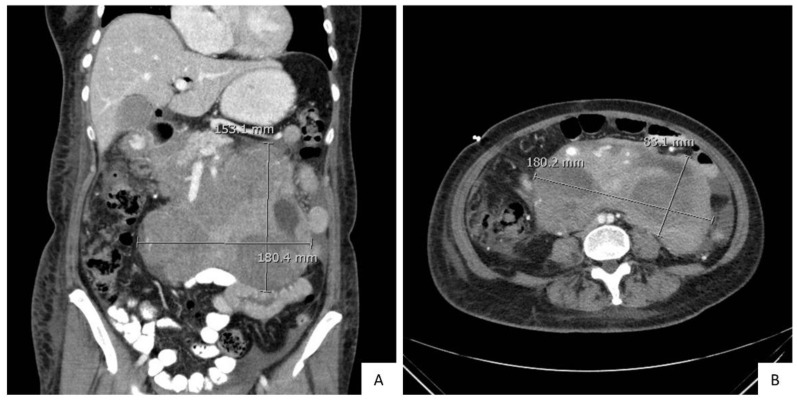
Comparison of abdominal and pelvic CT with contrast, post treatment. (**A**). Coronal, and (**B**). transverse planes show large lobulated and heterogeneously enhancing mass involving the head of the pancreas and retroperitoneum. Mass is decreased to 17 × 19 cm *versus* 19 × 22 cm (Figure 1A–C) after the initiation of treatment.

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
