# Peer review of "High-grade B-cell Lymphoma with MYC and BCL2 Rearrangement Arising from Follicular Lymphoma: Presentation as a Large Peripancreatic Mass"

_diagnostics, 2020, doi:10.3390/diagnostics10030157_

Round 1
Reviewer 1 Report
the paper reported a comprehensive clinical, pathologic and radiographic review of the patient with previously undiagnosed low-grade follicular lymphoma that transformed into a double-hit lymphoma. CT imaging and PET study were demonstrated carefully and summarized the process of diagnosis. the treatment for the patient is very valuable for similar patients. the paper is recommended to be published after minor revision.
"A 57-year-old Asian female, with a past medical history of hypertension, hypercholesterolemia and pre-diabetes, presented with abdominal distension and acute right upper quadrant pain." Dose the patient's medical history promote the transformation of "double-hit lymphoma". Could you offer negative control of H&E, CD20,CD10, BCL2, cmyc, Ki-67 staining to highlight the difference of B-cell of patient. I believe this situation of the patient is not an independent case, there should be other similar cases, could you check reference to compare with other published similar cases to show the difference of treatment.Author Response
Response to Reviewer 1 Comments
Point 1:
The paper reported a comprehensive clinical, pathologic and radiographic review of the patient with previously undiagnosed low-grade follicular lymphoma that transformed into a double-hit lymphoma. CT imaging and PET study were demonstrated carefully and summarized the process of diagnosis. the treatment for the patient is very valuable for similar patients. the paper is recommended to be published after minor revision.
"A 57-year-old Asian female, with a past medical history of hypertension, hypercholesterolemia and pre-diabetes, presented with abdominal distension and acute right upper quadrant pain." Dose the patient's medical history promote the transformation of "double-hit lymphoma".
Could you offer negative control of H&E, CD20,CD10, BCL2, cmyc, Ki-67 staining to highlight the difference of B-cell of patient.
Response 1: Please provide your response for Point 1.
I appreciate the request to include negative control to highlight neoplastic B-cells. If I understand correctly reviewer would like to see H&E and immunostains on the FNA of non-neoplastic/reactive B-cell proliferation. Unfortunately, we do not have an appropriate case to include as a Figure in this manuscript. I described normal findings in the text.
Lines 122-127: “In this setting differential diagnosis with reactive B-cell proliferation is not feasible due to the presence of sheets of highly atypical B-cells, that showed surface kappa-light chain restriction, and abnormal immunophenotype profile. Reactive B-cells express CD20, are expected to be smaller, and are usually admixed with T-cells. Germinal cell type non-neoplastic B-cells in addition to CD20 express CD10 and Bcl6, but do not express Bcl2 and c-myc. Finally, high proliferative index, estimated by Ki-67 at 85% is indicative of a high-grade B-cell neoplasm”.
Point 2:
I believe this situation of the patient is not an independent case, there should be other similar cases, could you check reference to compare with other published similar cases to show the difference of treatment.
Response 2:
Lines 200-203: Transformation to a “double hit” lymphoma has been well documented and can present a diagnostic challenge[18] [19] [20]. Patients that transform to “double hit” lymphoma usually require dose-adjusted etoposide, prednisone, vincristine, cyclophosphamide, doxorubicin and rituximab (EPOCH-R) [13-15].
Reviewer 2 Report
The authors describe a case of double-hit lymphoma with a concomitant diagnosis of low-grade lymphoma in an inguinal lymph node.
The case is presented very clearly and provides a good example of an excellent work-up of a case. Introduction and discussion are written well.
Major comments:
My main comment is a lack of novelty. It is well known that FL can transform into a DLBCL, and some carry the BCL-2 translocation and acquire a second MYC translocation. Furthermore, the clinical implication of revealing the FL in the inguinal node is minor, and would not affect treatment decision. The minimal spread of the low-grade lymphoma, argues against transformation. It could be that the patient developed both FL and DLBCL lymphoma. The way to address this question is to compare the B cell rearrangement (by PCR) between the DLBCL and FL.Taken together, while a well written, clear and fluent paper, I fell it does not contribute to the practicing hemato-oncologist.
Author Response
Response to Reviewer 2 Comments
Point 1:
My main comment is a lack of novelty. It is well known that FL can transform into a DLBCL, and some carry the BCL-2 translocation and acquire a second MYC translocation. Furthermore, the clinical implication of revealing the FL in the inguinal node is minor, and would not affect treatment decision. The minimal spread of the low-grade lymphoma, argues against transformation. It could be that the patient developed both FL and DLBCL lymphoma.
Response 1:
Lines 203-209: Aggressive EPOCH-R regimen can be potentially curative in the instances of de novo “double-hit” lymphomas. However, it is not effective in eradicating an underlying low-grade follicular lymphoma. While the major cause of the relapse is due to the high-grade component, in patients with the transformed follicular lymphoma, follicular lymphoma recurrence can occur [21]. In addition patients with underlying low-grade lymphoma are more likely to be resistant to Rituximab and radioimmunotherapy due to the possible prior treatment of the low-grade component [21] [22].
Point 2:
The way to address this question is to compare the B cell rearrangement (by PCR) between the DLBCL and FL.
Response 2:
We performed molecular clonality studies using next generation sequencing of the immunoglobulin variable region from peripancreatic and inguinal lymph node. The same clone V7-4 was detected.
Lines 81-85: In addition, molecular clonality studies confirmed the same origin of low-grade and high-grade components of lymphoma. The next generation sequencing of the immunoglobulin variable region determined the same clone, V7-4, in low-grade follicular lymphoma involving inguinal lymph node and high-grade lymphoma with BCL2/IGH and MYC rearrangements involving peripancreatic region.
Lines 163-167: In addition, clonality studies using next generation sequencing of the variable region of immunoglobulin gene identified identical clone, V7-4, in both inguinal lymph node involved by low-grade follicular lymphoma and peripancreatic high-grade B-cell lymphoma with BCL2 and MYC rearrangements. Therefore, diagnosis of a high-grade B-cell lymphoma with MYC and BCL2 rearrangements that transformed from a low-grade follicular lymphoma was rendered.
Lines 215-216: Molecular clonality studies might be instrumental to establish the same origin in cases of high-grade B-cell lymphoma that do not show concurrent low-grade component.
Round 2
Reviewer 2 Report
I thank the authors for completing the missing molecular analysis indeed confirming the high-grade lymphoma has evolved from the low-grade follicular lymphoma.
Again, the paper is well written, however, lack novelty